# Characterizing the Convergence of Game Dynamics via Potentialness

**Martin Bichler**                                            *m.bichler@tum.de*
*Department of Computer Science, Technical University of Munich*

**Davide Legacci**                       *davide.legacci@univ-grenoble-alpes.fr*
*Univ. Grenoble Alpes, CNRS, Inria, Grenoble INP, LIG*

**Panayotis Mertikopoulos**              *panayotis.mertikopoulos@imag.fr*
*Univ. Grenoble Alpes, CNRS, Inria, Grenoble INP, LIG*

**Matthias Oberlechner**                      *matthias.oberlechner@tum.de*
*Department of Computer Science, Technical University of Munich*

**Bary Pradelski**                                      *bary.pradelski@cnrs.fr*
*Maison Française d'Oxford, CNRS*

**Reviewed on OpenReview:** *https://openreview.net/forum?id=Is9APiPg4V*

## Abstract

Understanding the convergence landscape of multi-agent learning is a fundamental problem of great practical relevance in many applications of artificial intelligence and machine learning. While it is known that learning dynamics converge to Nash equilibrium in potential games, the behavior of dynamics in many important classes of games that do not admit a potential is poorly understood. To measure how "close" a game is to being potential, we consider a distance function, that we call "potentialness", and which relies on a strategic decomposition of games introduced by Candogan et al. (2011). We introduce a numerical framework enabling the computation of this metric, which we use to calculate the degree of "potentialness" in generic matrix games, as well as (non-generic) games that are important in economic applications, namely auctions and contests. Understanding learning in the latter games has become increasingly important due to the wide-spread automation of bidding and pricing with no-regret learning algorithms. We empirically show that potentialness decreases and concentrates with an increasing number of agents or actions; in addition, potentialness turns out to be a good predictor for the existence of pure Nash equilibria and the convergence of no-regret learning algorithms in matrix games. In particular, we observe that potentialness is very low for complete-information models of the all-pay auction where no pure Nash equilibrium exists, and much higher for Tullock contests, first-, and second-price auctions, explaining the success of learning in the latter. In the incomplete-information version of the all-pay auction, a pure Bayes-Nash equilibrium exists and it can be learned with gradient-based algorithms. Potentialness nicely characterizes these differences to the complete-information version.

## 1 Introduction

Multi-agent systems and multi-agent learning have drawn considerable attention, owing to their massive deployment in machine learning enabled applications and their increasing economic impact, with agents automatically ordering goods, setting prices, or bidding in auctions (Yang & Wang, 2020). In contrast to other applications of machine learning, the input in multi-agent learning is non-stationary and depends on the strategic behavior and learning of other agents, which leads to challenging computation and learning problems that go well beyond the "business as usual" framework of empirical risk minimization.

The literature on learning in games has a long history and asks what type of equilibrium behavior (if any) may arise in the long run of a process of learning and adaptation, in which agents are trying to (myopically) maximize their payoff while adapting to the actions of other agents through repeated interactions (Fudenberg & Levine, 1998; Hart & Mas-Colell, 2003). To that end, many learning algorithms have been developed ranging from iterative best-response to first-order online optimization algorithms in which agents follow their utility gradient in each step (Mertikopoulos & Zhou, 2019; Bichler et al., 2023).

It is known that the dynamics of learning agents do not always converge to a Nash equilibrium (NE) (Daskalakis et al., 2010; Flokas et al., 2020): they may cycle, diverge, or be chaotic, even in zero-sum games, where computing Nash equilibria is tractable (Mertikopoulos et al., 2018; Palaiopanos et al., 2017). In fact, (Hart & Mas-Colell, 2003) showed that in general uncoupled dynamics do not converge to Nash equilibrium in all games. With this negative result at hand, there lacks a comprehensive characterization of games that are "learnable"; however our understanding is much better for certain classes of games.

On the one hand, no-regret algorithms and dynamics do not converge in games with only mixed Nash equilibria (Flokas et al., 2020; Giannou et al., 2021a). On the other hand, if the underlying game is an exact potential game, then no-regret and other learning algorithms converge to an $\varepsilon$-NE with minimal exploration (Heliou et al., 2017; Mertikopoulos et al., 2024). Importantly, even though many classes of games of interest are not exact potential games (Candogan et al., 2013a;b), experimental evidence shows that even if games are not exact potential games, learning dynamics often converge to Nash equilibrium. Beyond exact potential games there is no good understanding of the behavior of learning dynamics.

Partially motivated by this, Candogan et al. (2011) introduced a game decomposition that allows one to characterize how "close" a game is to being potential by resolving it into a potential and a harmonic component (plus a "non-strategic" part which does not affect the game's equilibrium structure and unilateral payoff differences). In contrast to potential games, the exponential weight / replicator dynamics – perhaps the most widely studied no-regret dynamics – does not converge in any harmonic game, and instead exhibit a quasi-periodic behavior known as Poincaré recurrence (Legacci et al., 2024). Based on this dynamical dichotomy between potential and harmonic games, we consider a measure of *potentialness* and analyze generic normal-form games, as well as several classes of games steming in economic applications. We show that potentialness provides a useful indicator for both the existence of pure Nash equilibria and the convergence of no-regret algorithms. While the latter was already part of the motivation of Candogan et al. (2013a) the former is a novel connection emerging from our empirical exploration. In particular, we find that the average potentialness in random games decreases and concentrates on a value with increasing numbers of agents or actions: Games with a potentialness below 0.4 rarely exhibit convergence, while games with values larger than 0.6 mostly do. We also categorize specific games, such as Jordan's matching pennies game (Jordan, 1993), where learning dynamics are known not to converge (or, more precisely, to converge to a non-terminating cycle of best responses).

Economically motivated games such as auctions and contests have more structure in the payoffs. The analysis of these games is relevant today because pricing and bidding are increasingly being automated via learning agents. Learning agents are used to bid in display ad auctions, but they are also used by automated agents that set prices on online platforms such as Amazon (Chen et al., 2016). Whether we can expect the dynamics of such multi-agent interactions to converge to an equilibrium or exhibit inefficient price cycles or even chaos, is an economically important question. We find that the potentialness is very low for all-pay auctions and much higher for Tullock contests, first- and second-price auctions. Indeed, our experiments show that learning algorithms do not converge for all-pay auctions, but they do so for the other economic

games. The low potentialness of the all-pay auction also connects to the fact that it does not possess a pure Nash equilibrium.

In summary, we use the decomposition of Candogan et al. (2011) to define a metric that helps us characterize whether a game is learnable under standard no-regret learning dynamics, and to what extent. This is in stark contrast to a brute-force approach, where different initial conditions need to be explored. Our analysis of economic games is motivated by the observation that first-order no-regret dynamics - such as the replicator dynamics in particular - appear to converge in a wide array of classes of games with important applications, such as auctions and contests. Known sufficient conditions for convergence such as monotonicity or variational stability (Mertikopoulos & Zhou, 2019) are not even satisfied in simple economic games such as the first-price auction (Bichler et al., 2025). However, the notion of potentialness provides a crisp characterization of these differences as it does in simple normal-form games such as Matching Pennies and the Prisoner's Dilemma. Overall, potentialness provides us with a means to characterize convergence in the many games where known sufficient conditions appear to be too strong. This can help us analyze algorithmic markets where learning algorithms are increasingly used to submit bids or set prices (Bichler et al., 2024).

## 2    Related Literature

In this paper, we focus throughout on repeated normal-form games, where players move simultaneously and receive the payoffs as specified by the combination of actions played. The theory of learning in games examines what kind of equilibrium arises as a consequence of a process of learning and adaptation, in which agents are trying to maximize their payoff while learning about the actions of other agents in repeated games (Fudenberg & Levine, 1998). For example, fictitious play is a natural method by which agents iteratively search for a pure Nash equilibrium (NE) and play a best response to the empirical frequency of play of other players (Brown, 1951). Several algorithms have been proposed based on best or better response dynamics for finite and simultaneous-move games, ultimately leading to a vast corpus of literature (Abreu & Rubinstein, 1988; Hart & Mas-Colell, 2000; Fudenberg & Levine, 1998; Hart & Mas-Colell, 2003; Young, 2004), while more recent contributions draw on first-order online optimization methods such as online gradient descent or online mirror descent to study the question of convergence (Mertikopoulos & Zhou, 2019; Bichler et al., 2023).

Learning dynamics do not always converge to equilibrium (Daskalakis et al., 2010; Flokas et al., 2020). Learning algorithms can cycle, diverge, or be chaotic; even in zero-sum games, where the NE is tractable (Mertikopoulos et al., 2018; Palaiopanos et al., 2017). Sanders et al. (2018) argues that chaos is typical behavior for more general matrix games. Recent results have shown that learning dynamics do not converge in games with mixed Nash equilibria (Giannou et al., 2021a;b). On the positive side, Mertikopoulos & Zhou (2019) showed conditions for which no-external-regret learning algorithms result in a NE in finite games if they converge. However, in general, the dynamics of matrix games can be arbitrarily complex and hard to characterize (Andrade et al., 2021).

While there is no comprehensive characterization of games that are "learnable" and one cannot expect that uncoupled dynamics lead to NE in all games (Hart & Mas-Colell, 2003), there are some important results regarding the broad class of no-regret learning algorithms. One can distinguish between internal (or conditional) regret and a weaker version, called 'external (or unconditional) regret'. External regret compares the performance of an algorithm to the best single action in retrospect; internal regret, on the other hand, allows one to modify the online action sequence by changing every occurrence of a given action by an alternative action. For learning rules that satisfy the stronger no-internal regret condition, the empirical frequency of play converges to the game's set of correlated equilibria (Foster & Vohra, 1997; Hart & Mas-Colell, 2000). The set of correlated equilibria (CE) is a non-empty, convex polytope that contains the convex hull of the game's Nash equilibria. The coordination in CE can be implicit via the history of play (Foster & Vohra, 1997). On the other hand, algorithms that are no-external-regret learners converge by definition to the set of coarse correlated equilibria (CCE) in finite games (Foster & Vohra, 1997; Hart & Mas-Colell, 2000). This set, in turn, contains the set of CE such that we get $NE \subset CE \subset CCE$. In contrast to correlated equilibria, in a coarse correlated equilibrium, every player could be playing a strictly dominated strategy

for all time (Viossat & Zapechelnyuk, 2013), which makes CCE a fairly weak, non-rationalizable solution concept.

An important class of games in which a variety of learning algorithms converge to NE is that of *potential games* (Monderer & Shapley, 1996). In these games, the difference in any player's utility under a change in strategy is captured by the variation of a global potential function; as a consequence, any sequence of improvements by players converges to a pure NE (Heliou et al., 2017; Christodoulou et al., 2012; Anagnostides et al., 2022). Remarkably, the class of congestion games is equivalent to that of potential games (Rosenthal, 1973; Monderer & Shapley, 1996); however, while many games are not potential games, no-regret algorithms still converge to NE, raising the following question: *what fundamental property enables this convergence?*

Partially motivated by this question, Candogan et al. (2011) propose a decomposition of finite normal form games into three components, with distinctive strategical properties. This decomposition can be leveraged to approximate a given game with a potential game, which in turn can be used to characterize the long-term behavior of the dynamics in the original game (Candogan et al., 2013a;b). In particular, Candogan et al. (2013a) examine the convergence of best-response and logit-best-response dynamics – in the sense of Blum & Kalai (1999) – and they show that, if only one player updates per turn, the dynamics remain convergent in slight perturbations of potential games.[1]

An important caveat is that the class of dynamics considered by Candogan et al. (2013a) can result in positive regret. Moreover, unlike the setting studied here, players do not move simultaneously but sequentially. Our focus is broader, as we aim to understand the behavior of no-regret dynamics across the full spectrum of potentialness – not just in near-potential games – and to determine where the convergence of regularized no-regret learning methods fails. In doing so, we also provide a first positive answer to the open question posed by Candogan et al. (2013a), who asked whether the replicator/follow-the-regularized-leader dynamics, a staple among no-regret learning schemes, remains convergent under small perturbations of potential games.

## 3 Preliminaries

In this section, we recall the definitions of normal-form game, Nash equilibrium, and potential game.

**Definition 1** (Normal-form game). *A finite normal-form game consists of a tuple $G = (\mathcal{N}, \mathcal{A}, u)$, where*

- *$\mathcal{N}$ is a finite set of $N$ players, indexed by $i$;*

- *$\mathcal{A} = \mathcal{A}_1 \times \cdots \times \mathcal{A}_N$, where $\mathcal{A}_i$ is a finite set of $A_i$ actions available to player $i$;*

- *$u = (u_1, \ldots, u_N)$, where $u_i : \mathcal{A} \mapsto \mathbb{R}$ is a* payoff *or* utility *function for player $i \in \mathcal{N}$.*

*An element $a = (a_1, \ldots, a_N) \in \mathcal{A}$ is referred to as an* action profile*; we denote by $A = \prod_i A_i$ the total number thereof.*

A normal-form game is hence fully specified by the prescription of the strategies available to each player, and of the outcomes resulting from a simultaneous and strategic interaction.

Rather than explicitly selecting an action – also called a *pure strategy* – from the available options, players may instead *mix*, meaning they randomize over their available actions according to a probability distribution, referred to as a *mixed strategy*.

**Definition 2** (Mixed strategy). *Given the normal-form game $G = (\mathcal{N}, \mathcal{A}, u)$, the set of* mixed strategies *for player $i$ is $S_i = \Delta(\mathcal{A}_i)$, the set of probability distributions (or lotteries) over $\mathcal{A}_i$. An element $s \in S$ is called* strategy profile*.*

A Nash equilibrium is a strategy profile such that no player has an incentive to unilaterally deviate from their chosen strategy:

---

[1] Importantly, the dynamics considered by Candogan et al. (2013a) are not the simultaneous best-reply dynamics of Gilboa & Matsui (1991) or the logit dynamics of Fudenberg & Levine (1999); Hofbauer & Sandholm (2009), but rather turn-by-turn updates where each player observes the play of their opponents and plays a (logit) best-response.

**Definition 3** (Nash equilibrium)**.** *In a normal-form game* $G = (\mathcal{N}, \mathcal{A}, u)$, *a strategy profile* $s^* = (s_1^*, s_2^*, \ldots, s_N^*) \in S_1 \times \ldots \times S_N$ *is a* Nash equilibrium *if*

$$u_i(s_i^*, s_{-i}^*) \geq u_i(s_i, s_{-i}^*) \quad \textit{for all } i \in \mathcal{N}, \ s_i \in S_i, \tag{1}$$

*where* $s_{-i}^* = (s_1^*, s_2^*, \ldots, s_{i-1}^*, s_{i+1}^*, \ldots, s_N^*)$.

A *potential game* is a game in which strategic interactions are fully captured by single a scalar function, aligning individual players' incentives with its maximization:

**Definition 4** (Potential game)**.** *A game* $G = (\mathcal{N}, \mathcal{A}, u)$ *is a called* potential *if there exists a function* $\phi : \mathcal{A} \to \mathbb{R}$ *such that, for every player* $i \in \mathcal{N}$ *and every pair of action profiles* $a, a' \in \mathcal{A}$ *that differ only in the action of player* $i$ *(that is,* $a_i \neq a_i'$ *and* $a_{-i} = a_{-i}'$), *the following condition holds:*

$$u_i(a) - u_i(a') = \phi(a) - \phi(a'). \tag{2}$$

*When this is the case,* $\phi$ *is called* potential function *of the game.*

## 4 Potentialness of a game

In this section we provide an overview of a combinatorial decomposition technique for finite games in normal form, originally introduced by Candogan et al. (2011). Subsequently, we leverage such decomposition to define the *potentialness* of a game, a measure of its closeness to being a potential game. This measure, closely related to the *maximum pairwise difference* introduced by Candogan et al. (2013a), can be used as a predictor for the existence of strict pure Nash-equilibria (SPNE) in a game, and of the limiting behavior of learning dynamics thereof.

**Deviation map** Given a finite game in normal form $G = (\mathcal{N}, \mathcal{A}, u)$, pairs of strategy profiles $(a, a')$ that differ only in the strategy of one player are called *unilateral deviations*, and their space is denoted by $\mathcal{E}$. Representing a game in terms of *utility differences between unilateral deviations* rather than in terms of utilities themselves captures the strategic structure of a game in an effective way, in the sense that games with different utilities but identical utility differences between unilateral deviations share the same set of Nash equilibria Candogan et al. (2011).

To achieve this representation of the game $G = (\mathcal{N}, \mathcal{A}, u)$, consider its *response graph* $\Gamma(\mathcal{N}, \mathcal{A})$, that is the graph with a node for each of the $A$ pure strategy profile in $\mathcal{A}$, and an edge for each of the $E := |\mathcal{E}| = \frac{A}{2} \sum_{i \in \mathcal{N}} (A_i - 1)$ unilateral deviations in $\mathcal{E}$ (Biggar & Shames, 2023).

This graph is an instance of a *simplicial complex* $K$, that is, loosely speaking, a collection of oriented $k$-dimensional faces (points, segments, triangles, tetrahedrons, ...) with $k \in \{0, 1, \ldots\}$ (Jonsson, 2007). Given a simplicial complex $K$ one can build a family of vector spaces $\{C_k\}_{k=0,1,\ldots}$, called *chain groups*, where each chain group $C_k$ is the space spanned by the $k$-dimensional faces of the complex $K$, i.e., the space of assignments of a real number to each $k$-dimensional face of the complex (Munkres, 1984).

In this work, we restrict our attention to the chain groups $C_0$ and $C_1$ on the response graph $\Gamma(\mathcal{N}, \mathcal{A})$ of a game. $C_0 \cong \mathbb{R}^A$ is the space of assignments of a real number to each vertex $a \in \mathcal{A}$, and $C_1 \cong \mathbb{R}^E$ is the space of assignments of a real number to each edge $(a, a') \in \mathcal{E}$; an element in $C_1$ is called *flow* on the graph.

Note that the potential function $\phi : \mathcal{A} \to \mathbb{R}$ of a potential game is precisely an assignment of a number to each pure strategy profile $a \in \mathcal{A}$: as such, it is an element of $C_0$. In a similar way, observe that the collection of utility functions $u = (u_i)_{i \in \mathcal{N}} : \mathcal{A} \to \mathbb{R}^N$ is the assignment of a number $u_i(a) \in \mathbb{R}$ to each pure strategy profile $a \in \mathcal{A}$ for each player $i \in \mathcal{N}$; as such, it can be considered as an element of "$N$ copies of $C_0$", that is $u \in \mathcal{U} := C_0 \times \cdots \times C_0$.

The key observation is that the differences between unilateral deviations of a game $G = (\mathcal{N}, \mathcal{A}, u)$ can be represented as a special flow $Du$ in $C_1$ on $\Gamma(\mathcal{N}, \mathcal{A})$, called *deviation flow of the game*, by means of the *deviation map*:

**Definition 5** (Deviation map). *The* deviation map *of the game* $G = (\mathcal{N}, \mathcal{A}, u)$ *is the linear map*

$$
\begin{aligned}
D : \mathcal{U} &\to C_1 \\
u &\longmapsto Du
\end{aligned}
\tag{3}
$$

*such that, for all* $u \in \mathcal{U}$ *and all* $(a, a') \in \mathcal{E}$

$$
(Du)_{(a,a')} = u_i(a') - u_i(a)
\tag{4}
$$

*for the (necessarily existing and unique)* $i \in \mathcal{N}$ *such that* $a_i \neq a'_i$.

In words, the deviation flow $Du \in C_1$ assigns to each edge $(a, a') \in \mathcal{E}$ of the response graph, i.e., to every unilateral deviation of the game, the utility difference of the deviating player $i \in \mathcal{N}$.

The image of the deviation map, $\operatorname{Im} D \subset C_1$, defines the space of *feasible flows* on a game's response graph. Among these, two types are particularly relevant: *potential* flows and *harmonic* flows.

**Potential flows** For a given number of players and actions per player, representing a game via its deviation flow $Du$ rather then its payoff $u$ preserves all the strategic information of the game, allowing for a concise characterization of potential games. To introduce it, we need the following definition:

**Definition 6** (Gradient map). *The* gradient map[2] *is the linear map*

$$
\begin{aligned}
d_0 : C_0 &\to C_1 \\
\phi &\longmapsto d_0\phi
\end{aligned}
\tag{5}
$$

*such that*

$$
(d_0\phi)_{(a,a')} = \phi(a') - \phi(a)
\tag{6}
$$

*for all* $\phi \in C_0$ *and all* $(a, a') \in \mathcal{E}$.

It is now immediate to show that

**Proposition 1.** *A game* $G = (\mathcal{N}, \mathcal{A}, u)$ *is potential with potential function* $\phi$ *if and only if* $Du = d_0\phi$ *for some* $\phi \in C_0$.

*Proof.* Let $G = (\mathcal{N}, \mathcal{A}, u)$ be a potential game with potential function $\phi$. Then

$$
Du_{(a,a')} = u_i(a') - u_i(a) = \phi(a') - \phi(a) = (d_0\phi)_{(a,a')}
$$

for all $(a, a') \in \mathcal{E}$, where $i \in \mathcal{N}$ is the actor of the deviation $(a, a')$. Thus, $Du = d_0\phi$. Conversely, let $G = (\mathcal{N}, \mathcal{A}, u)$ be a game with $Du = d_0\phi$ for some $\phi \in C_0$. Then

$$
u_i(a') - u_i(a) = Du_{(a,a')} = (d_0\phi)_{(a,a')} = \phi(a') - \phi(a)
$$

for all $i \in \mathcal{N}$ and all $(a, a') \in \mathcal{E}$ acted by $i$. Thus, the game is potential with potential function $\phi$. $\square$

The proposition means that the space of potential games is the linear subspace $D^{-1} \operatorname{Im} d_0 \subset \mathcal{U}$; in light of this we call the image of the gradient map, $\operatorname{Im} d_0 \subset C_1$, the space of *potential flows*.

**Harmonic flows** Endowing $C_0$ and $C_1$ with an Euclidean-like inner product $\langle \cdot, \cdot \rangle_k$ one can define the *divergence map* $\partial_1 := d_0^\dagger : C_1 \to C_0$ as the adjoint operator of the gradient map, namely $\langle X, d_0\phi \rangle_1 = \langle \partial_1 X, \phi \rangle_0$ for all $X \in C_1$ and all $\phi \in C_0$. The flows in the subspace $\ker \partial_1 \in C_1$ are called *harmonic flows*[3], and the games whose flow is harmonic, i.e., the games in $D^{-1} \ker \partial_1 \subset \mathcal{U}$, are called *harmonic games*.

---

[2]The gradient map is an instance of so-called *co-boundary maps*; see Munkres (1984) for details.

[3]The term *harmonic* refers in combinatorial Hodge theory Dodziuk (1976) to the kernel of the *Laplacian operator* $\ker \Delta_1 = \ker \partial_1 \cap \ker d_1$, where $d_1$ is defined analogously to $d_0$. As Candogan et al. (2011) show, each feasible flow belongs to $\ker d_1$, making these two definitions of harmonic flows consistent.

**Hodge decomposition of feasible flows**   Leveraging the *combinatorial Hodge decomposition theorem*[4] it can be shown that potential flows and harmonic flows completely characterize feasible flows:

**Theorem** (Candogan et al. (2011) — Combinatorial Hodge decomposition of feasible flows). *The space of feasible flows is the orthogonal direct sum of the subspaces of potential flows and harmonic flows:*

$$\operatorname{Im} D = \operatorname{Im} d_0 \oplus \ker \partial_1 \tag{7}$$

Equivalently, every feasible $Du$ flow can be decomposed in a unique way as $Du = Du_p + Du_h$, where the potential flow $Du_p \in \operatorname{Im} d_0$ is the orthogonal projection of $Du$ onto $\operatorname{Im} d_0$, and the harmonic flow $Du_h \in \ker \partial_1$ is the orthogonal projection of $Du$ onto $\ker \partial_1$.

The decomposition (7) takes place in the space $\operatorname{Im} D \subset C_1$ of feasible flows. In Appendix A.1, we discuss how to derive a corresponding decomposition in the space $\mathcal{U}$ of actual payoffs, allowing for the unique decomposition of any game's payoff $u \in \mathcal{U}$ as $u = u_{\mathcal{P}} + u_{\mathcal{H}} + u_{\mathcal{K}}$, where $u_{\mathcal{P}}$ is a *normalized potential* game, $u_{\mathcal{H}}$ a *normalized harmonic* game, and $u_{\mathcal{K}}$ a *non-strategic* game; c.f. Figure 1 for an example, and Candogan et al. (2011) for further details. However, the decomposition (7) in the space of feasible flows suffices to introduce the measure of potentialness used in this work, as the deviation flow of non-strategic games is identically zero (see Appendix A.1).

Figure 1: Decomposition of the Shapley game.

Endowing $\mathcal{U}$ with an inner product structure, Candogan et al. (2011) show that $u_{\mathcal{P}}$ is the potential game *closest* to $u$. This allows Candogan et al. (2013a;b) to characterize the limiting behavior of dynamics in the game $u$ in terms of the properties of the potential game $u_{\mathcal{P}}$ and of the *distance* between $u$ and $u_{\mathcal{P}}$, a concept that is made precise in the next paragraph.

**Potentialness**   The potential component $Du_p$ of the deviation flow $Du$ of a game can be used to build a measure of how close to being a potential game the game is. To compute it, Candogan et al. (2011) introduce the orthogonal projection onto the subspace of potential flows; by the properties of the Moore-Penrose pseudo-inverse $\tilde{d}_0 : C_1 \to C_0$ of the gradient map (Golan, 1992) such projection is $e := d_0 \tilde{d}_0 : C_1 \to C_1$, so that

$$Du_p = eDu \in \operatorname{Im} d_0 \subset C_1 \tag{8}$$

Candogan et al. (2013a;b) use the deviation map to define the *maximum pairwise difference* between two games as $\delta(u, u') = \|Du - Du'\|$.[5] In particular, since the potential component $u_{\mathcal{P}}$ of a game $u$ is (up to a non-strategic game) the potential game closest to the original game, they use the maximum pairwise difference $\delta(u, u_{\mathcal{P}}) = \|Du - Du_p\| = \|Du_h\|$ between a game $u$ and its potential component $u_{\mathcal{P}}$ as a measure of closeness to being a potential game for the game $u$. In this spirit we give the following definition:

---

[4]See Jiang et al. (2011) for a concise presentation and proof.
[5]In this work we use the 2-norm, whereas Candogan et al. (2013a) use the infinity norm.

**Definition 7** (Potentialness). *The potentialness of a game $G = (\mathcal{N}, \mathcal{A}, u)$ is the real number*

$$P(u) := \frac{||Du_p||}{||Du_p|| + ||Du_h||} \tag{9}$$

**Proposition 2.** *The potentialness of a game fulfills*

*(i)* $P(u) \in [0, 1]$

*(ii)* $P(u) = 1 \iff \delta(u, u_{\mathcal{P}}) = 0 \iff u$ *is a potential game*

*(iii)* $P(u) = 0 \iff u$ *is a harmonic game*

*Proof.* Consider the game $G = (\mathcal{N}, \mathcal{A}, u)$ and its potentialness

$$P(u) := \frac{||Du_p||}{||Du_p|| + ||Du_h||}$$

(i) It is obvious that $P(u) \in [0, 1]$;

(ii) Since $\delta(u, u_{\mathcal{P}}) = ||Du_h||$ it is also obvious that $P(u) = 1 \iff \delta(u, u_{\mathcal{P}}) = 0$. Let this be the case, then $Du = Du_p + Du_h = Du_p \in \operatorname{Im} d_0$, so $Du = d_0\phi$ for some $\phi \in C_0$, i.e. the game is potential by Proposition 1.

Conversely, let $G = (\mathcal{N}, \mathcal{A}, u)$ be a potential game. Then $Du_p = Du$, since $Du_p$ is the orthogonal projection of $Du$ onto $\operatorname{Im} d_0$, and such projection leaves $Du$ invariant since $Du = d_0\phi$ itself belongs to $\operatorname{Im} d_0$. Hence, $||Du_h|| = 0$.

(iii) It is obvious that $P(u) = 0 \iff ||Du_p|| = 0$; if this is the case then $Du = Du_h$, so the game is harmonic by definition.

Conversely, if the game is harmonic then $Du = Du_h$ by an argument analogous to the one in point (ii), which implies that $||Du_p|| = 0$. $\qquad\square$

In light of these properties, a game's potentialness provides a concise measure of its proximity to being a potential game. In the next sections, we examine the computational complexity of computing potentialness, and explore the existence of strict pure Nash equilibria (SPNE) as well as the asymptotic behavior of learning dynamics as a function of potentialness.

**Scalability** To compute the potentialness of a game with payoff functions $u \in \mathcal{U}$, two computationally expensive steps are required: computing the deviation flow, $u \mapsto Du$, and projecting it onto the space of potential flows, $Du \mapsto eDu$. Since these operations involve the linear maps $D : \mathcal{U} \to C_1$ of Equation (3) and $e : C_1 \to C_1$ of Equation (8), the calculations reduce to matrix-vector multiplications.

Given a game $G = (\mathcal{N}, \mathcal{A}, u)$, the operators $D$ and $e$ depend only on the number of actions $A$ and agents $N$, not on the payoff function $u \in \mathcal{U}$. Thus, the matrices representing these operators need to be computed only once for each combination of $N$ and $A$, allowing them to be precomputed and reused. However, this preprocessing step is costly, as matrix size grows exponentially with the number of agents. The projection $Du \mapsto eDu$, the most computationally expensive step, involves the matrix $e$ of size $\dim C_1 \times \dim C_1$, where $\dim C_1 = \frac{A}{2} \sum_{i \in \mathcal{N}} (A_i - 1)$. In games where all agents have the same number of actions, i.e., $A_i = m$ for all $i \in \mathcal{N}$, the number of entries in $e$ scales as $\mathcal{O}(N^2 m^{2N+2})$. For reference, in a game with five actions per agent, the number of entries in $e$ is of order $10^4$ for $N = 2$, $10^5$ for $N = 3$, and $10^7$ for $N = 4$.

For the larger settings considered in Figure 2 (three agents with 12 actions each, and four agents with 7 actions each), computing these matrices takes approximately 2–3 minutes. However, this cost is incurred only once. Once the matrices are available, computing potentialness is very fast, as the dominant cost reduces to a matrix-vector multiplication of complexity $\mathcal{O}(N^2 m^{2N+2})$. The average runtime (over 100 runs) for computing potentialness in our experiments, performed on a standard notebook, is shown in Figure 2.

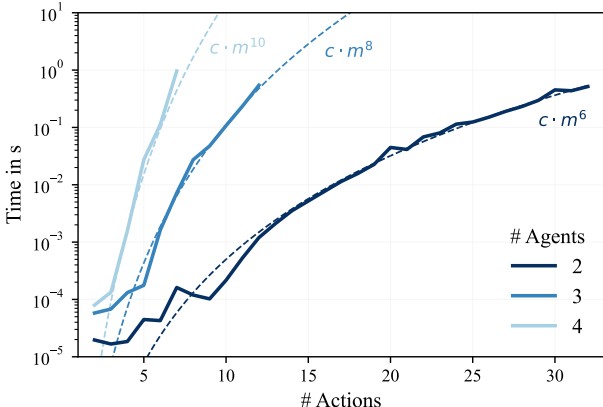

Figure 2: Average runtime for computing potentialness over 100 runs, with precomputed matrices.

## 5 Numerical Experiments

We conduct our numerical experiments on randomly generated and economically motivated games. Our analysis focuses mainly on two aspects, namely, the existence of strict pure Nash-equilibria (SPNE) and the convergence of learning dynamics as a function of the potentialness of the games.[6]

To analyze learning dynamics, we consider online mirror descent (OMD) (Nemirovskij & Yudin, 1983) with entropic regularization, leading to the update steps outlined in Algorithm 1. OMD is a natural choice among no-regret algorithms due to its favorable regret properties, and belongs to the broader class of follow-the-regularized-leader (FTRL) algorithms (Shalev-Shwartz, 2012).

---

**Algorithm 1:** Online Mirror Descent

**Input** : initial mixed strategies $s_{i,0}$
**for** $t = 1$ **to** $T$ **do**
    **for** *agent* $i = 1$ **to** $N$ **do**
        observe gradient $v_{i,t}$;
        set $s_{i,t} \leftarrow \mathcal{P}_{s_{i,t-1}}(\eta_t v_{i,t})$;
    **end**
**end**

---

The entropic regularization introduces in Algorithm 1 a *prox-mapping* $\mathcal{P}_x : \mathbb{R}^d \to \mathbb{R}^d$ of the form

$$\mathcal{P}x(y) = \frac{(x_j \exp(y_j))j = 1^d}{\sum_{j=1}^d x_j \exp(y_j)}, \tag{10}$$

where $d \in \mathbb{N}$ denotes the dimension of the player's strategy space, i.e., $d = A_i$; see, e.g., Nemirovski (2005); Juditsky et al. (2011) for further details.

The convergence properties of the algorithm depend on the chosen step-size $\eta_t$. We use a step-size sequence of the form $\eta_t = \eta_0 \cdot t^{-\beta}$ for some $\eta_0 > 0$ and $\beta \in (0, 1]$, and consider Algorithm 1 to have converged to an (approximate) NE of the game if the *relative utility loss*, $\ell_i(s_t) = 1 - u_i(s_t)/u_i(br_i, s_{-i,t})$, falls below a predefined tolerance of $\varepsilon = 10^{-8}$ for all agents $i \in \mathcal{N}$ within a fixed number of iterations, $T = 2\,000$. In the above, $br_i$ denotes a best response of agent $i$ given the opponents' strategy profile $s_{-i,t}$.

---

[6]The code is available on Github at `https://github.com/MOberlechner/games_learning`.

**Standard Games** Before examining randomly generated games, we first briefly analyze the potentialness of standard games. The results are summarized in Table 1.

| Game | Actions | $P(u)$ |
|---|---|---|
| Matching Pennies | 2x2 | 0.00 |
| Battle of the Sexes | 2x2 | 0.94 |
| Prisoners' Dilemma | 2x2 | 1.00 |
| Shapley Game | 3x3 | 0.36 |
| Jordan Game $(\alpha, \beta)$ | 2x2 | [0.00, 0.50] |

Table 1: Potentialness of standard matrix games. The payoff matrices for the first three games are from Nisan (2007), while the matrix for the Shapley game is given in Figure 1. The matrix for the Jordan game is taken from Jordan (1993, Def 2.1). OMD converges to a pure NE only in the Battle of the Sexes and the Prisoner's Dilemma, both of which exhibit relatively high values of potentialness.

Pure equilibria exist only in the *Battle of the Sexes* and the *Prisoner's Dilemma*. The Prisoner's Dilemma, being a potential game, has a potentialness of exactly 1, whereas the Battle of the Sexes, despite not being a potential game, still exhibits relatively high potentialness. In both cases, OMD converges to a pure NE.

*Matching Pennies*, the *Shapley Game*, and the *Jordan Game* are known to have only mixed equilibria. Since OMD can converge only to strict NE (Giannou et al., 2021a), it does not converge in these games. Consistently, Matching Pennies, as a harmonic game, has a potentialness of exactly zero, while the Shapley and Jordan games, being neither harmonic nor potential, exhibit relatively low potentialness.

Interestingly, the potentialness of the Jordan Game, when the parameters of the payoff matrices are sampled uniformly at random, varies between 0 and 0.5.

**Random Games** In this section, we investigate three key aspects of potentialness in random games. First, we analyze *how potentialness varies* as a function of the number of agents and available actions. Second, we explore the connection between potentialness and the existence of *strict Nash equilibria*. Third, we examine the relationship between potentialness and the *long-term behavior of learning dynamics* in random games.

To generate a random game $G = (\mathcal{N}, \mathcal{A}, u)$ for a certain *setting* (number of agents $N$ and actions per agent $A_i$), we independently sample each payoff value from a uniform distribution, i.e., $u_i(a) \sim \mathcal{U}([0,1])$ for all $a \in \mathcal{A}$ and $i \in \mathcal{N}$; our dataset consists of $10^6$ games for each considered setting.

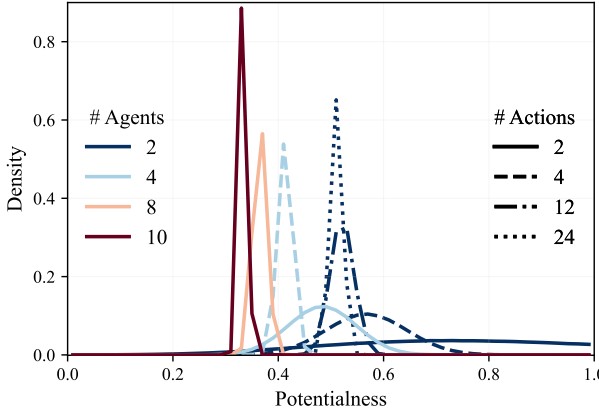

Figure 3: Empirical distribution of potentialness in $10^6$ randomly generated games for each considered setting. Increasing the size of the games reduces the variance and the mean of the distribution.

The resulting empirical distributions of potentialness for randomly generated games with different settings are presented in Figure 3; this leads us to the following observation:

**Observation 1.** *As the size of the game increases (in terms of either the number of agents or actions), both the variance and the mean of the observed potentialness distribution decrease.*

Our next objective is to analyze the connection between potentialness and the long-term behavior of learning dynamics. As a natural intermediate step, we first examine the existence of *strict* Nash equilibria – strategy distributions for which Equation (1) holds as a strict inequality – as a function of potentialness. The reason is twofold. First, strict Nash equilibria are the only possible candidates for the convergence of learning algorithms: Giannou et al. (2021a) show that a strategy distribution is asymptotically stable under FTRL algorithms (such as Algorithm 1) if and only if it is a strict Nash equilibrium. Second, strict Nash equilibria necessarily correspond to *pure* strategy profiles. Since potential games always admit at least one pure Nash equilibrium (Monderer & Shapley, 1996), one might expect that games with high levels of potentialness have a higher probability of possessing a strict (hence, pure) Nash equilibrium (SPNE).[7]

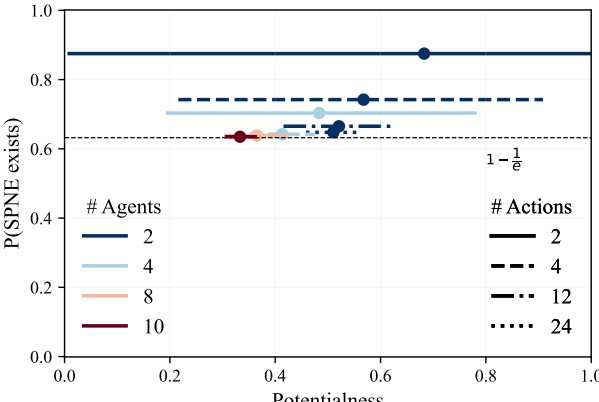

Figure 4: Relationship between potentialness and existence of SPNE *across different settings.* Vertical axis: empirical probability of SPNE existence as a function of the setting. Horizontal axis: Potentialness values. The width of the horizontal lines represents the observed range of potentialness values for each setting, corresponding to the support of the densities shown in Figure 3. The dot on each line indicates the average potentialness for that setting. As game size increases, mean potentialness decreases, and the probability of SPNE existence approaches $1 - 1/e$ (dotted horizontal line).

Rinott & Scarsini (2000) showed that the probability of the existence of a pure Nash equilibrium in randomly drawn games converges to $1-1/e$ as either the number of players or the number of actions per player increases. This tendency is evident in Figure 4, which also illustrates the following: *across different settings*, as the average potentialness increases, so does the probability of SPNE existence.

Next, we examine the relationship between SPNE existence and potentialness *within the same setting.* For each setting, we discretize the potentialness range $[0, 1]$ into 20 intervals $I_k := (\frac{k-1}{20}, \frac{k}{20}]$ for $k = 1, \ldots, 20$, partition games accordingly, and compute the fraction of games in each group that have at least one SPNE; the results are visualized in Figure 5. We summarize our findings in the following observation:

**Observation 2.** *Both across different settings and within the same setting, a higher potentialness increases the likelihood that a game has at least one SPNE.*

---

[7]A pure Nash equilibrium fails to be strict if ties occur. In the context of random games, a pure Nash equilibrium is strict with probability 1, as ties occur with probability zero.

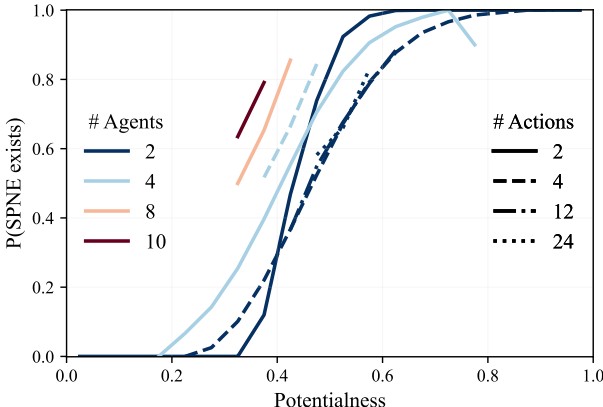

Figure 5: Relationship between potentialness and existence of SPNE *within the same setting*. Vertical axis: empirical probability of SPNE existence as a function of potentialness. Horizontal axis: potentialness values. For each setting, a higher potentialness increases the likelihood that a game has at least one SPNE.

Next, we examine the relationship between potentialness and the long-term behavior of learning dynamics in random games. The existence of an SPNE ensures *local* convergence of learning algorithms (Mertikopoulos & Zhou, 2019), prompting the question of whether higher potentialness not only increases the probability of having an SPNE, but also influences its *basin of attraction* – the set of initial conditions leading to convergence.

Following the same approach as before, we partition for each setting (number of agents and actions) the potentialness range into 20 intervals, $I_k := (\frac{k-1}{20}, \frac{k}{20}]$ for $k = 1, \ldots, 20$, and group together games with potentialness in $I_k$. For each group $k$, we run Algorithm 1 with step-size $\eta_t = \eta_0 \cdot t^{-\beta}$, letting $\eta_0 = 2^3$ and $\beta = 20^{-1}$. The initial condition is fixed to the uniform strategy, $s_{i,0} = A_i^{-1}\mathbf{1}$, for all agents $i$. The fraction of converging instances in games that possess a SPNE is plotted against the group's potentialness value, with results shown in Figure 6 and summarized in Observation 3.

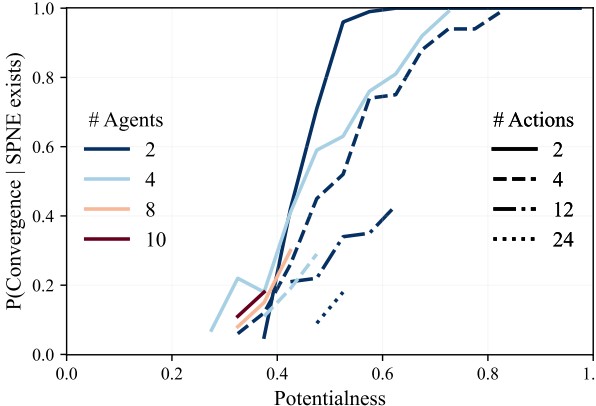

Figure 6: Empirical probability of convergence of Algorithm 1 as a function of potentialness. The analysis considers only random games with at least one SPNE; higher potentialness increases the likelihood of reaching equilibrium.

**Observation 3.** *Higher potentialness increases the likelihood of reaching equilibrium using Algorithm 1, assuming the existence of at least one SPNE.*

**Remark.** *Different initial conditions, uniformly sampled over the feasible space, lead to the same outcomes as stated in Observation 3, as discussed in Appendix A.2. Employing smaller step-sizes similarly leads to analogue results; however, bigger step-sizes ensure better convergence rates.*

**Economic Games**  As compared to random matrix games, many games analyzed in the economic sciences have more structure in the payoffs. Arguably, one of the most important classes of economic games are auctions and contests, which are widely used to describe strategic interaction in markets (Krishna, 2009). In these games, the utility functions of agents have a specific form, and the allocation rule is monotonic in the bids. Well-known auctions and contests include:

- First-Price Sealed-Bid (FPSB) Auction

$$u_i(a, v_i) = x_i(a) \cdot (v_i - a_i) \tag{11}$$

- Second-Price Sealed-Bid (SPSB) Auction

$$u_i(a, v_i) = x_i(a) \cdot (v_i - \max_{j \neq i} a_j) \tag{12}$$

- All-Pay Auction

$$u_i(a, v_i) = x_i(a) \cdot v_i - a_i \tag{13}$$

- War of Attrition (WoA)

$$u_i(a, v_i) = x_i(a) \cdot (v_i - \max_{j \neq i} a_j) - (1 - x_i(a)) \cdot a_i \tag{14}$$

- Tullock Contest

$$u_i(a, v_i) = \begin{cases} v \cdot \frac{a_i}{\Sigma_j a_j} - a_i & \text{if } \Sigma_j a_j > 0 \\ \frac{v}{n} & \text{else} \end{cases} \tag{15}$$

In the above, $a_i \in \mathcal{A}_i$ represents the *bid* of agent $i$, and $a \in \mathcal{A} = \prod_{i \in \mathcal{N}} \mathcal{A}_i$ denotes the complete bid (or action) profile. Similarly, $v_i$ corresponds to agent $i$'s *value*, and $x_i(a) : \mathcal{A} \to [0, 1]$ is the *allocation function*, which incorporates a random tie-breaking rule, given by

$$x_i(a) = \begin{cases} \frac{1}{n_{\max}} & \text{if } a_i = \max_{j \in \mathcal{N}} a_j, \\ 0 & \text{if } a_i < \max_{j \in \mathcal{N}} a_j, \end{cases} \tag{16}$$

where $n_{\max}$ denotes the number of bids that attain the maximum.

Auctions are typically defined over continuous action spaces, where each player's bid is a real number, normalized without loss of generality to the interval $\mathcal{A}_i = [0, 1]$. However, the decomposition proposed by Candogan et al. (2011) applies only to finite games. To make it applicable, we discretize each action space $\mathcal{A}_i$ into $A_i$ equidistant points. We remark that, for practical purposes, auctions *are* inherently discrete, as bids can only be submitted up to a finite number of decimal places.

The choice of discretization appears to have little impact on a game's potentialness. Figure 7 presents potentialness levels computed for varying discretizations and agents valuations. As the discretization becomes finer, the potentialness stabilizes at an asymptotic value; however, the number of pure Nash equilibria can exhibit sharp changes. For instance, in the symmetric First-Price Sealed-Bid Auction, there is one strict NE for $A_i = 21$, whereas for $A_i = 20$, an additional non-strict pure NE emerges alongside the strict NE.

**Observation 4.** *The potentialness of the discretized economic games given in Equations* (11) *to* (15) *is an inherent property of the underlying (continuous) game, and does not depend on the choice of discretization.*

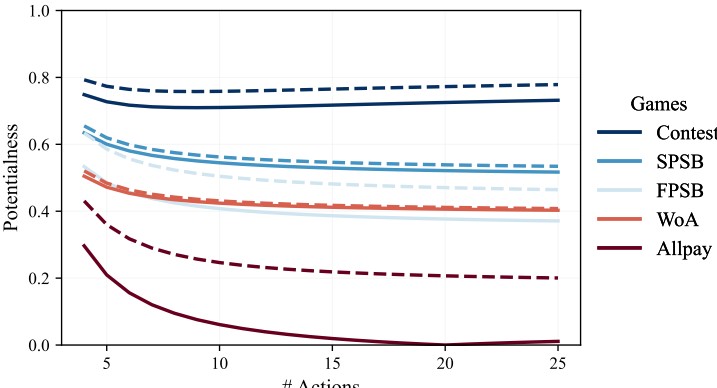

Figure 7: Potentialness of the economic games given in Equations (11) to (15), with 2 agents. We consider different discretizations (that is, number of actions), and different valuations. The solid lines show the potentialness in the symmetric setting, where both agents have values $v_1 = v_2 = 1$, while the dashed line shows the asymmetric settings with $v_1 = \frac{3}{4}, v_2 = 1$. As the discretization becomes finer, the potentialness stabilizes at an asymptotic value.

A detailed analysis of the decomposition of first- and second-price auctions reveals that their harmonic components are identical. When varying the payment rule through convex combinations of the first- and second-price rules, only the potential and non-strategic components of the decomposition change. This observation supports the intuition that the allocation rule determines the strategic difficulty posed by the game. In contrast, contests, that have a smoothed version of this allocation, show higher level of potentialness.

Given a game decomposed into its potential and harmonic components, $u = u_{\mathcal{P}} + u_{\mathcal{H}}$, potentialness describes the relative weight of the potential component over the harmonic one. One can then build a game with prescribed level of potentialness by considering the convex combination $u_\alpha := \alpha u_{\mathcal{P}} + (1 - \alpha)u_{\mathcal{H}}$, with $\alpha \in [0, 1]$. Figure 8 shows the convergence behavior of Algorithm 1 as a function of potentialness in games built with this procedure from the economic games given in Equations (11) to (15):

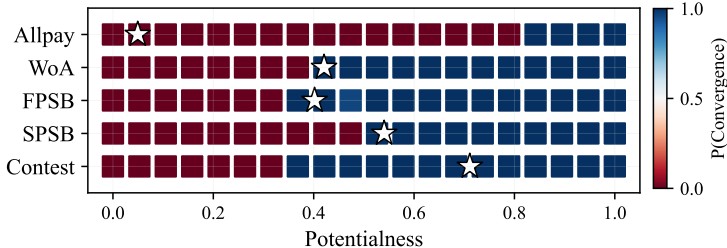

Figure 8: Convergence of OMD in economic games. Each square corresponds to a convex combination $u_\alpha := \alpha u_{\mathcal{P}} + (1 - \alpha)u_{\mathcal{H}}$ of the potential and harmonic components of the respective economic game (vertical axis); the corresponding level of potentialness can be read on the horizontal axis. The color of the square shows the empirical frequency of convergence to an equilibrium using OMD ($\eta = 2^8, \beta = \frac{1}{20}$) for $10^2$ randomly generated initial strategies. The stars indicate the potentialness of the original economic game. For each game, there exists a potentialness threshold above which the algorithm converges; this threshold is lower for games with a higher original potentialness, suggesting that convergence in games with high potentialness is relatively more robust against harmonic perturbations.

We observe that, for each game, there exists a *potentialness threshold* beyond which the algorithm begins to converge. As expected, this threshold coincides with the existence of an SPNE: When potentialness exceeds a certain game-specific value, an SPNE exists, and and Algorithm 1 converges; below this threshold, no pure equilibrium exists, and Algorithm 1 fails to converge. Furthermore, this threshold is lower for games with a higher original potentialness, suggesting that convergence in games with high potentialness is relatively more robust against harmonic perturbations.

Among the considered economic games, the all-pay auction is the "hardest to learn", namely it is the one with lowest potentialness, and hightest convergence threshold. Interestingly, while the all-pay auction in the complete-information setting only has a mixed equilibrium, its *incomplete-information* counterpart can admit pure *Bayes-Nash equilibria* (BNE). The work of Bichler et al. (2023) shows that BNE are learnable in discrete games using first-order methods such as Algorithm 1 . To better understand the different equilibrium structures between the complete and the incomplete information case, we extend our analysis to the richer Bayesian framework, examining it through the lens of potentialness.

**Bayesian Economic Games**  *Bayesian* (or *incomplete-information*) games are the standard approach in auction theory to model the economic games we considered (Krishna, 2009). Compared to a normal-form game, a Bayesian games $B = (\mathcal{N}, \mathcal{V}, \mathcal{A}, F, u)$ is characterized by an additional *type space* $\mathcal{V}$ and a known *prior distribution* $F$ over this type space. Each player $i \in \mathcal{N}$ observes a private type, which is drawn from $\mathcal{V}_i$ according to the prior distribution $F$. In the examples given in Equations (11) to (15), the private type is the valuation $v_i$. After observing their values, the agents play an action, i.e., submit a bid $a_i \in \mathcal{A}_i$, and receive their utility $u_i(a_i, a_{-i}, v_i)$, which depends on their type. A pure strategy in such a Bayesian game is a function $\beta_i : \mathcal{V}_i \to \mathcal{A}_i$ that maps players' types to actions. Given a strategy profile, one can compute the expected (ex-ante) utility $\tilde{u}_i = \mathbb{E}_{v \sim F}[u_i(\beta_i(v_i), \beta_{-i}(v_{-i}), v_i)]$. A *Bayes-Nash equilibrium* (BNE) is a strategy profile $\beta$, where no player can increase the expected utility $\tilde{u}$ by deviating.

As in the complete-information case, we consider only finite action spaces, and additionally assume finite type spaces. This allows us to reformulate the Bayesian game as a finite normal-form game, decompose it using Equation (7), and analyze its potentialness, as given by Equation (9). The actions of the induced normal-form game consist of all the possible strategies of the original incomplete-information game. Specifically, an agent with $V_i$ types and $A_i$ actions in the Bayesian game translates to an agent with $A_i^{V_i}$ pure actions in the normal-form representation. Clearly, the size of the resulting normal-form game grows rapidly and quickly becomes intractable. To mitigate this issue, we restrict our analysis to *non-decreasing strategies*, as is customary in auction theory. This reduces the number of strategies to $\binom{V_i + A_i - 1}{A_i - 1}$, enabling us to analyze Bayesian games with two players and up to four actions and types, corresponding to 35 non-decreasing strategies. The payoffs in the normal-form game represent the expected utilities in the Bayesian game, and the pure Nash equilibria in the normal-form game correspond to Bayes-Nash equilibria the Bayesian game.

Following this procedure, we constructed the normal-form representation of Bayesian economic games with four actions, up to four types, and a uniform prior distribution. Our findings, summarized in Figure 9 and Observation 5, reveal a notable pattern: the Bayesian version of each game exhibits *higher potentialness* than its complete-information counterpart (where the number of types is one). Specifically, potentialness increases as the number of types grows.

In particular, the increase in potentialness is substantial for the all-pay auction; with two or four types, it even leads to the existence of a pure BNE. This is especially noteworthy because learning algorithms successfully converge to equilibrium in Bayesian games with a BNE (Bichler et al., 2023), whereas in the complete-information version of the all-pay auction – which only admits mixed Nash equilibria – convergence does not occur, cf. Figure 8.

**Observation 5.** *Potentialness consistently increases with the number of types in the Bayesian versions of the considered games. In particular, while Algorithm 1 fails to converge in the complete-information all-pay auction, it successfully converges to a pure Bayes-Nash equilibrium in the Bayesian version of the game.*

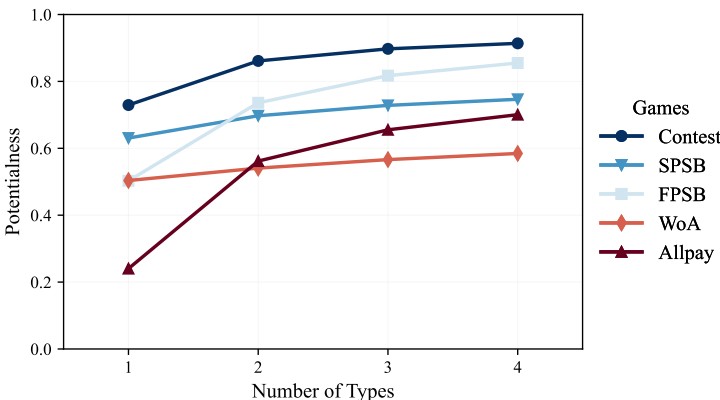

Figure 9: Potentialness of Bayesian economic games. We consider the Bayesian version of economic games with two agents, each having four actions, $\mathcal{A}_i = 0.0, 0.3, 0.6, 0.9$, and independent, uniformly distributed types $\mathcal{V}_i = \{\frac{i}{V_i} \mid i = 1, \ldots, V_i\}$, where $V_i = 4$ denotes the number of types. Potentialness consistently increases as the number of types grows.

## 6 Conclusions

Understanding whether learning algorithms converge to a Nash equilibrium has been a long-standing question in game theory, and characterizing which games permit convergence and which do not remains an open problem in the study of learning in games. While it is well established that exact potential games guarantee convergence, we extend this analysis by relaxing this condition, investigating convergence in games that are quantitatively distant from being potential games. A decomposition technique allows us to measure how "potential" a specific normal-form game is: For random matrix games, we find that potentialness serves as a strong predictor for the existence of strict Nash equilibria, and for the convergence of online mirror descent. Economically motivated games, such as auctions and contests, impose additional structure on payoffs. Here, too, potentialness – observed to increase with the number of types – proves to be a reliable predictor of convergence.

Unlike individual simulation runs, where convergence heavily depends on algorithm initialization, potentialness provides an ex-ante predictor of average convergence, offering a systematic approach to studying learning in games. This insight addresses a long-standing question in the literature by shifting the focus from empirical simulations to a more principled measure of convergence potential.

Future research should aim to further validate these findings by examining a wider range of no-regret learning dynamics and expanding the analysis to broader classes of games. Additionally, improving the computational efficiency of potentialness, extending its application to continuous games, and leveraging it to design more effective learning algorithms remain key open directions.

**Acknowledgments**

This project was funded by the Deutsche Forschungsgemeinschaft (DFG, German Research Foundation) under Grant No. BI 1057/9; and Project Number 277991500. It is based upon work supported by the National Science Foundation under Grant No. DMS-1928930 and by the Alfred P. Sloan Foundation under grant G-2021-16778, while M. Bichler, D. Legacci, B. Pradelski, and M. Oberlechner were in residence at the Simons Laufer Mathematical Sciences Institute (formerly MSRI) in Berkeley, California, during the Fall 2023 semester. This work was also supported in part by the French National Research Agency (ANR) in the framework of the PEPR IA FOUNDRY project (ANR-23-PEIA-0003), the "Investissements d'avenir" program (ANR-15-IDEX-02), the LabEx PERSYVAL (ANR-11-LABX-0025-01), MIAI@Grenoble Alpes (ANR-19-P3IA-0003) and IRGA-SPICE. P. Mertikopoulos is also affiliated with Archimedes, Athena Research Center,

and was partially supported by project MIS 5154714 of the National Recovery and Resilience Plan Greece 2.0 funded by the European Union under the NextGenerationEU Program.

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

## A   Appendix

### A.1   Decomposition in the space of payoffs

The reason Candogan et al. (2011) work in the space $C_1$ of flows rather than in the space $\mathcal{U}$ of payoffs to achieve a decomposition of games is twofold: first, the combinatorial Hodge theorem holds true on the space $C_1$ of flows (and in general on every chain group $C_k$ of a simplicial complex), while there is no such a theorem for the space $\mathcal{U}$ of payoffs; second, two games $u \neq u'$ such that $Du = Du'$, albeit having different payoffs, display the same strategical properties and are effectively the "same" game[8], so looking at the deviation flow rather than at the payoff one gets rid of a redundancy that is intrinsic in the formulation of a game.

**Non-strategic games**   To make this last point more precise Candogan et al. (2011) introduce the notion of *non-strategic games.*

**Definition 8.** *A finite game in normal form $G = (\mathcal{N}, \mathcal{A}, u)$ is called non-strategic if it has zero deviation flow:*

$$Du = 0 \,. \tag{17}$$

*The space of non-strategic games is denote by*

$$\mathcal{K} := \ker D \,. \tag{18}$$

In a non-strategic game, all players are indifferent among all of their choices:

**Proposition 3** (Candogan et al. 2011)**.** *The game $G = (\mathcal{N}, \mathcal{A}, u)$ is non-strategic if and only if*

$$u_i(a'_i, a_{-i}) = u_i(a''_i, a_{-i}) \,, \tag{19}$$

*for all $i \in \mathcal{N}$, all $a_{-i} \in \mathcal{A}_{-i}$, and all $a'_i, a''_i \in \mathcal{A}_i$.*

Since $D : \mathcal{U} \to C_1$ is a linear map between vector spaces, the space of non-strategic games is a linear subspace $\mathcal{K} \subset \mathcal{U}$. It follows by the definition of non-strategic games that two games have the same deviation flow if and only if their difference is a non-strategic game, and in this case we say that the two games are *strategically equivalent.*

**Normalized games**   Being strategically equivalent is an equivalence relation on the space $\mathcal{U}$ of payoffs; one can select a representative element in each equivalence class $[u]$ by choosing a complement of $\mathcal{K}$ in $\mathcal{U}$ and projecting $u \in \mathcal{U}$ onto such complement along $\mathcal{K}$. A natural choice is that of using the *orthogonal* complement $\mathcal{K}^\perp$ of the space of non-strategic games with respect to the Euclidean inner-product in $\mathcal{U}$; we refer to such procedure as *normalization*, and following Candogan et al. (2011) we give the following definition:

**Definition 9.** *A finite game in normal form $G = (\mathcal{N}, \mathcal{A}, u)$ is called normalized if*

$$u \in \mathcal{K}^\perp \,. \tag{20}$$

Normalized games enjoy the "no-leftover" property: the sum of any player's payoffs over their choices is zero for any fixed choice by the other players.

**Proposition 4** (Candogan et al. 2011)**.** *The game $G = (\mathcal{N}, \mathcal{A}, u)$ is normalized if and only if*

$$\sum_{a'_i \in \mathcal{A}_i} u(a'_i, a_{-i}) = 0 \tag{21}$$

*for all $i \in \mathcal{N}$ and all $a_{-i} \in \mathcal{A}_{-i}$.*

---

[8]Quoting Candogan et al. (2013b), *if [two games have the same deviation flow], then the equilibrium sets of these games are identical. However, payoffs at equilibria may differ, and hence they may be different in terms of their efficiency (such as Pareto efficiency) properties (see Candogan et al. (2011)).*

**Decomposition in the space of payoffs**  After normalizing the space $\mathcal{U}$ of payoffs[9] one can translate the decomposition of feasible flows (Equation (7) in the main text) from $\operatorname{Im} D \subset C_1$ to $\mathcal{U}$ by means of the *Moore-Penrose pseudo-inverse* $\tilde{D} : C_1 \to \mathcal{U}$ of the deviation map.

Recall from Section 4 that the space of potential games is $D^{-1} \operatorname{Im} d_0 \subset \mathcal{U}$, and that the space of harmonic games is $D^{-1} \ker \partial_1 \subset \mathcal{U}$. Their intersections with the space $\mathcal{K}^\perp \subset \mathcal{U}$ of normalized games give the spaces of *normalized potential games* and of *normalized harmonic games*:

**Definition 10.** *The space of normalized potential games is the linear subspace*

$$\mathcal{P} := \textit{(potential games)} \cap \mathcal{K}^\perp \subset \mathcal{U} . \tag{22}$$

*The space of normalized harmonic games is the linear subspace*

$$\mathcal{H} := \textit{(harmonic games)} \cap \mathcal{K}^\perp \subset \mathcal{U} . \tag{23}$$

**Theorem** (Candogan et al. (2011) — Combinatorial Hodge decomposition of finite normal form games)**.** *The space $\mathcal{U}$ is the direct sum of the subspaces of normalized potential games, normalized harmonic games, and non-strategic games:*

$$\mathcal{U} = \mathcal{P} \oplus \mathcal{H} \oplus \mathcal{K} . \tag{24}$$

*Equivalently, given a finite normal form game $G = (\mathcal{N}, \mathcal{A}, u)$ the payoff function $u$ can be uniquely decomposed as the sum $u = u_\mathcal{P} + u_\mathcal{H} + u_\mathcal{K}$ of a normalized potential game $u_\mathcal{P}$, a normalized harmonic game $u_\mathcal{H}$, and a non-strategic game $u_\mathcal{K}$.*

**Decomposition components**  Recall that the deviation map is a linear map $D : \mathcal{U} \to C_1$ from the space of payoffs to the space of flows. By the properties of the Moore-Penrose pseudo-inverse $\tilde{D} : C_1 \to \mathcal{U}$ of the deviation map (Golan, 1992), the operator $\Pi := \tilde{D}D : \mathcal{U} \to \mathcal{U}$ is the orthogonal projection onto $\mathcal{K}^\perp = (\ker D)^\perp$. Recall furthermore that $e : C_1 \to C_1$ is the orthogonal projection onto the subspace of potential flows.

These operator can be used to obtained explicit expressions for the components of the decomposition in the space of payoffs:

**Proposition 5** (Candogan et al. (2011))**.** *Given the finite normal form game $G = (\mathcal{N}, \mathcal{A}, u)$ the components $u_\mathcal{P}, u_\mathcal{H}$ and $u_\mathcal{K}$ of the combinatorial Hodge decomposition of finite normal form games are given by*

- $u_\mathcal{K} = u - \Pi u \in \mathcal{K}$   ;

- $u_\mathcal{P} = \tilde{D}eDu \in \mathcal{P}$   ;

- $u_\mathcal{H} = u - u_\mathcal{K} - u_\mathcal{P} \in \mathcal{H}$   .

---

[9]That is, after quotienting away the kernel of the deviation map.

## A.2 Additional Numerical Experiments

Following the experiment illustrated in Figure 6, we grouped together games with similar potentialness, and applied Algorithm 1 with $\eta_0 = 2^3$ and $\beta = 1/20$. However, instead of fixing the uniform strategy as the initial condition, we randomly sampled 25 initial strategies for each game. The first plot in Figure 10 shows the empirical probability of convergence of Algorithm 1 as a function of potentialness in various settings. The findings align with those of Figure 6, confirming that higher potentialness increases the likelihood of reaching equilibrium – regardless of specific initializations.

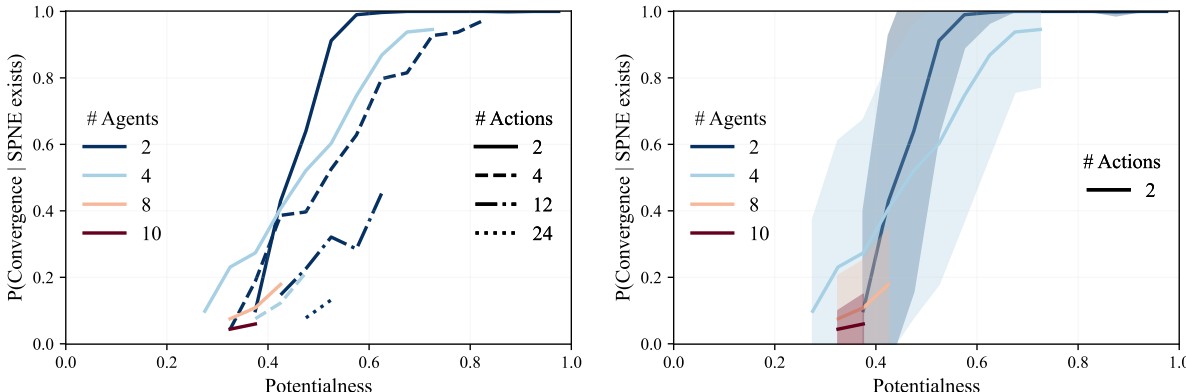

Figure 10: Empirical probability of convergence of Algorithm 1 in random games with SPNE as a function of potentialness in various settings. Left: randomly drawn initialization points. Right: standard deviation of the convergence probability.

On the second plot, we restrict our focus to settings with only 2 actions, and include the standard deviation of the convergence probability (colored areas) along with empirical probability of convergence (solid lines) of Algorithm 1 as a function of potentialness. We observe that the probability of convergence given a strict NE – i.e., the size of the equilibrium's basin of attraction – varies significantly among games with similar potentialness, as reflected in the relatively large standard deviations. However, the overall relationship between high potentialness and high convergence remains robust in expectation.

