# OpenReview forum: "Characterizing the Convergence of Game Dynamics via Potentialness"
_TMLR — Accepted by TMLR_

### Review · Reviewer_HBmG · 2024-10-05

**Summary Of Contributions:**

This paper introduces the concept of "potentialness" as a measure to assess how close a game is to being a potential game, which is crucial for predicting the convergence of multi-agent learning systems. The authors provide empirical evidence that potentialness is indicative of both the existence of pure Nash equilibria and the convergence of no-regret learning algorithms. Furthermore, the paper investigates the impact of the number of agents and actions on potentialness and compares various economic games using this metric.

**Audience:**

Yes

**Broader Impact Concerns:**

I think there are no related ethical concerns about this paper.

**Claims And Evidence:**

No

**Requested Changes:**

**Critical changes that would impact recommendation for acceptance:**

The paper could benefit from improving the overall writing quality. Some sections lack clarity and could be made more concise. I recommend a thorough revision to enhance readability and ensure the arguments are presented more clearly. Authors should add the correct code link to verify the implementability.

**Changes that would strengthen the work:**

Additionally, the authors should further elaborate on the applicability and feasibility of the proposed algorithm in large-scale games and in settings with continuous action spaces. This would provide a more comprehensive understanding of the algorithm's scalability and practical utility in more complex scenarios.

**Strengths And Weaknesses:**

**Strengths:**

The concept of potentialness is novel and offers a unique perspective for analyzing game convergence. The paper clearly distinguishes its contributions from prior work and provides appropriate citations to related literature. While I am not an expert in game theory, I have reviewed the mathematical derivations to the best of my ability, and they appear to be correct.

**Weaknesses:**

It would be beneficial for the authors to explore how the concept of potentialness scales to more complex game structures, such as those involving continuous action spaces.
The paper would benefit from a more detailed discussion of the computational complexity associated with calculating potentialness in large-scale games.
There are some typographical errors: "adaptto" should be corrected to "adapt to" (page 1), there are errors in the references (pages 1-2), and the code link provided is invalid (page 8).

---

> ### Author Response · Authors · 2024-12-17
> **Rebuttal by Authors**
>
> Dear Reviewer HBmG,
>
> Thank you for your input and your time. For your convenience, we reproduce and reply below to your questions one-by-one (grouping relevant comments together to keep the thread as concise and easy-to-read as possible).
>
> > It would be beneficial for the authors to explore how the concept of potentialness scales to more complex game structures, such as those involving continuous action spaces. The paper would benefit from a more detailed discussion of the computational complexity associated with calculating potentialness in large-scale games.
> > [...]
> > Additionally, the authors should further elaborate on the applicability and feasibility of the proposed algorithm in large-scale games and in settings with continuous action spaces. This would provide a more comprehensive understanding of the algorithm's scalability and practical utility in more complex scenarios.
>
> These questions are intertwined, so we treat them as one.
>
> First, regarding games with continuous action spaces, it should be noted that there is no known Hodge decomposition that could be leveraged to define the "potentialness" of a continuous game. While Hodge / de Rham theory could be a viable approach to eventually obtain such a decomposition, no such theory exists for the moment, so it is not feasible to define the potentialness of a game "directly".
>
> Instead, a fruitful - and natural - way to define the potentialness of a continuous game would be by means of its discretization. This is straightforward when the discretization mesh does not lead to a size explosion of the discretized game - but, otherwise, there is, of course, a scalability issue.
>
> In this regard, the problem of computing the potentialness of a game boils down to two basic steps: (1) computing and storing two large matrices; and (2) performing two matrix-vector multiplications. Assuming that all of the $N$ players have the same number of actions, $m$, these matrices are of order $\mathcal{O}(N^2 m^{2N+2})$. As discussed in [CMOP13], this leads to a complexity that is polynomial in the number of strategies but exponential in the number of players. The problem can be recast as a linear program [CMOP13], but the complexity remains the same (though the use of interior-point methods could greatly speed up the computation in practice). In all cases, the complexity of computing the potentialness of a game is a low-degree polynomial (with degree determined by the matrix multiplication constant) in the number of bits required to repesent the game.
>
> We did not go into these details in our submission because we wanted to focus on the relation between potentialness and learnability - thanks for giving us the opportunity to comment on this!
>
> > There are some typographical errors: "adaptto" should be corrected to "adapt to" (page 1), there are errors in the references (pages 1-2), and the code link provided is invalid (page 8).
>
> Thank you for your careful proofreading and for pointing out these minor glitches; all fixed, thanks again for bringing them to our attention. As for the link, we omitted the correct link in our original submission to preserve anonymity, and we opted to submit the code in our paper's supplementary material. We will of course provide the correct link and publish the code after the review process.
>
> > The paper could benefit from improving the overall writing quality. Some sections lack clarity and could be made more concise. I recommend a thorough revision to enhance readability and ensure the arguments are presented more clearly. Authors should add the correct code link to verify the implementability.
>
> Point taken, we will make sure to eliminate any remaining typos / unclear points. As for the link, we omitted the correct link in our original submission to preserve anonymity, and we opted to submit the code in our paper's supplementary material. We will of course provide the correct link and publish the code after the review process.
>
>
> Thank you again for your input. Kind regards,
>
> The authors

---

### Review · Reviewer_4yCV · 2024-10-25

**Summary Of Contributions:**

The submission investigates a metric of multi-player games, named _potentialness_, that correlates with (1) the existence of pure NE and (2) the convergence of no-regret algorithms. The authors lead with a detailed description of the preliminaries on potential games and related algorithms. The authors motivated the definition of potentialness via game decomposition, and evaluated it on random games, economic games, as well as some classical examples. Through these experiments, the authors empirically demonstrate the behavior of potentialness as #players and #actions vary, and its correlation with convergences of various algorithms. In particular, for continuous games, the authors also evaluated the effect of discretization for computing potentialness.

**Audience:**

Yes

**Claims And Evidence:**

Yes

**Requested Changes:**

Small changes:
1. intro paragraph 2: adaptto -> "adapt to"
2. intro paragraph 3: missing bibtex (shows up as question mark)

**Strengths And Weaknesses:**

The manuscript is well-motivated and clearly written. The definition of potentialness is intuitive, and appears to well-correlate with algorithm convergence. The paper provided a comprehensive study on the proposed metric, including the complexity of an algorithm that computes it, as well as behaviour under various games (continuous, different # players, etc.).

I have a few questions and comments:
1. The values of potentialness seem only comparable among similar #players and #actions, as shown in Figures 5 and 6. Have the authors considered variants that are more invariant to these parameters (so that, e.g., we can compare games with different #players or #actions)?
2. I have trouble understanding Figure 4. What are the lengths of the horizontal lines supposed to indicate?

---

> ### Author Response · Authors · 2024-12-17
> **Rebuttal by Authors**
>
> Dear Reviewer 4yCV,
>
> Thank you for your input and your time. For your convenience, we reproduce and reply below to your questions one-by-one (collecting related comments together to keep the thread as concise and easy-to-read as possible).
>
> > 1. The values of potentialness seem only comparable among similar #players and #actions, as shown in Figures 5 and 6. Have the authors considered variants that are more invariant to these parameters (so that, e.g., we can compare games with different #players or #actions)?
>
> Indeed, comparing the potentialness of games of different size and dimensionality can be quite challenging in general, as the values of the potentialness vary significantly with the number of actions or agents. However, the variation in randomly generated games is to be expected,  as it is more likely to draw a potential game in a smaller setting, like a 2x2 game, than in the case of games with a significantly larger number of players.
>
> Nevertheless, even in this case, the notion of potentialness can be a useful measure of comparison. For example, in games with an inherent structure - e.g.,those that arise in economic applications, such as auctions, oligopoly models and the like - the size of the game as determined by e.g., the discretization of the original economic model has little impact on the potentialness. This suggests that potentialness is an inherent property of the underlying game itself and not of the number of actions (cf. Observation 4 in our paper).
>
> > 2. I have trouble understanding Figure 4. What are the lengths of the horizontal lines supposed to indicate?
>
> Thanks for pointing out the lack of clarity at this point! To address it, we added explicit descriptions of our paper's visualizations; in Fig. 4 in particular, the lines in question indicate the level of potentialness observed in the specific setting, i.e., the support of the densities as seen in Figure 3.
>
> > 3. [Minor changes]
>
> Thank you for your careful proofreading and for pointing out these minor glitches; all fixed, thanks again for bringing them to our attention.
>
>
> Thank you again for your input. Kind regards,
>
> The authors

---

### Review · Reviewer_gADN · 2024-12-02

**Summary Of Contributions:**

The paper introduces the property of potentialness (in the sense of [1]) as a way to (empirically) characterise a game in terms of existence of strict pure NE and the limiting behaviour of several learning dynamics for several games of interests.


[1] Ozan Candogan, Ishai Menache, Asuman Ozdaglar, and Pablo A Parrilo. Flows and decompositions of games: Harmonic and potential games. Mathematics of Operations Research, 36(3):474–503, 2011.

**Audience:**

Yes

**Broader Impact Concerns:**

No concerns

**Claims And Evidence:**

Yes

**Requested Changes:**

Out of one error in references visualised with "(??)", I would recommend to address at least two of the following points in order to increase the contributions of the paper:

- Extend the analysis to more complex games, for example state-based (Markov) games.
- Analyse possible ways to measure potentialness in a numerically efficient way.
- Try to add some insights on how standard algorithms should be adapted in view of this property.

In general, what I am more concerned about is that the contributions seem limited to some empirical observations over some cases of interests, based on a (numerically challenging) property directly inspired on previous work, which I think would not be sufficient for the standards.

**Strengths And Weaknesses:**

**Strength**:

the paper addresses a problem of interest, namely the extent to which being near to a potential game is a useful metric to the tractability of a game, several works are starting to address the problem in even more complex scenarios than the ones considered here, for instance [1] addresses Markov Games.

**Weaknesses**:

the paper contributions are rather limited to my view. The definition is based directly on previous works and the theoretical characterisation is thus the one of related works, the numerical challenges of the which are not addressed (i.e. the numerical burden is described, rather then treated), the definition is descriptive and not prescriptive (no insights on algorithmic design are available, such as for [2] for instance).

[1] An $\alpha$-potential game framework for $N$-player dynamic games, Xin Guo and Xinyu Li and Yufei Zhang
[2] Newton Optimization on Helmholtz Decomposition for Continuous Games Giorgia Ramponi, Marcello Restelli

---

> ### Author Response · Authors · 2024-12-17
> **Rebuttal by Authors**
>
> Dear Reviewer gADN,
>
> Thank you for your input and your time. For your convenience, we reproduce and reply below to your questions one-by-one (collecting related comments together to keep the thread as concise and easy-to-read as possible).
>
> > The paper introduces the property of potentialness (in the sense of [CMOP11]) as a way to (empirically) characterise a game in terms of existence of strict pure NE and the limiting behaviour of several learning dynamics for several games of interests.
> > [...]
> > The definition is based directly on previous works and the theoretical characterisation is thus the one of related works.
>
> We would first like to clarify that the notion of potentialness did not appear in the paper of Candogan et al. [CMOP11], which focused on establishing the fundamental potential-harmonic decomposition and exploring its ramifications for the equilibrium structure of the game. It can be argued that [CMOP11] and some of its follow-ups ultimately wanted to leverage this decomposition to study the asymptotic behavior of best-reply dynamics in near-potential games, but the notion of potentialness and its impact on the replicator dynamics was neither introduced nor explored in [CMOP11].
>
> The goal of our paper was to use the decomposition of [CMOP11] to define a concrete metric that would help us characterize whether a game is learnable under standard no-regret learning dynamics, and to what extent. This was in turn motivated by the fact that first-order no-regret dynamics - such as the replicator dynamics in particular - appear to converge in a wide array of classes of games with important applications, such as auctions and contests. These games exhibit very different equilibrium properties: for example, while Tullock contests, first- and second-price auctions have pure Nash equilibria and appear to be learnable for the most part, all-pay auctions do not, so convential wisdom would suggest that such auctions are not learnable. The notion of potentialness provides a crisp characterization of these differences as it does in simple normal-form games such as Matching Pennies and the Prisonner's Dilemma.
>
> > The numerical challenges of [computing the potentialness] are not addressed (i.e. the numerical burden is described, rather then treated)
> > [...]
> > - Analyse possible ways to measure potentialness in a numerically efficient way.
>
> Excellent question, thanks for raising it.
>
> As noted in our submission, the problem of computing potentialness reduces to two key steps: (1) computing and storing two large matrices and (2) performing two matrix-vector multiplications. Assuming that all of the $N$ players have the same number of actions, $m$, these matrices are of order $\mathcal{O}(N^2 m^{2N+2})$. As discussed in [CMOP13], this leads to a complexity that is polynomial in the number of strategies but exponential in the number of players. The problem can be recast as a linear program [CMOP13], but the complexity remains the same (though the use of interior-point methods could greatly speed up the computation in practice). In all cases, the complexity of computing the potentialness of a game is a low-degree polynomial (with degree determined by the matrix multiplication constant) in the number of bits required to repesent the game.
>
> We did not go into these details in our submission because we wanted to focus on the relation between potentialness and learnability - thanks for giving us the opportunity to comment on this!

---

> > ### Author Response · Authors · 2024-12-17
> > **Rebuttal by Authors (2)**
> >
> > > the definition is descriptive and not prescriptive (no insights on algorithmic design are available, such as for [RR20] for instance).
> > > [...]
> > > - Try to add some insights on how standard algorithms should be adapted in view of this property.
> >
> > We understand your point, but there is a fundamental difference between the setting of [RR20] and our own. As in the original paper by Balduzzi et al. [BRMF+18], the setting considered in [RR20] concerns an *unconstrained* continuous game, and the authors leverage the decomposition of the player's individual payoff gradient field into a symmetric and skew-symmetric component to introduce a "symplectic adjustment" in the gradient ascent dynamics (with [RR20] altering the Hamiltonian adjustment of [BRMF+18] through the inclusion of a Newton-type preconditioner).
> >
> > The operative difference here is that the dynamics of [BRMF+18] evolve over an *unconstrained* vector space; this is also the case for [RR20] which treats stochastic game policies, not in their tabular representation (that is, as points in a product of simplices), but in a parametrized setting, with the parameters taking values in an unconstrained space.
> >
> > In a constrained setting - such as learning in finite games, which takes place over the simplex - the symplectic / Helmholtz decomposition cannot be applied as the dynamics do not take a simple gradient form. In fact, to undertake a Helmholtz decomposition that is compatible with, e.g., the replicator dynamics in this setting, one would have to endow the game's (constrained) strategy space with a suitable Riemannian metric, and then define the various components - gradients and symplectic forms - in a way that is compatible with said metric. To the best of our knowledge, the only step that has been taken in this direction is the very recent paper of Legacci et al. [LMP24], which endows the game with the Shahshahani metric and then performs a Helmholtz-Hodge decomposition based on this metric. However, even in the case of [LMP24], it is not at all clear how to use a Helmholtz decomposition of finite games in order to introduce a symplectic adjustment to the players' learning dynamics.
> >
> > Thus, while we agree that the use of potentialness in a prescriptive way would be an important direction to pursue, this would first require significant advances in our understanding of constrained game dynamics, and it is not something that can be undertaken in the current paper.
> >
> > > I would recommend to address at least two of the following points in order to increase the contributions of the paper:
> > >
> > > - Extend the analysis to more complex games, for example state-based (Markov) games.
> >
> > While an extension to more complex games is, of course, a very fruitful research direction, we would like to argue that there is still a lot to be learned and understood from a more detailed analysis of competititive scenarios that can be modeled as finite games. Such games already model several real-world applications that are of central interest to economics and machine learning, from resource allocation to display-ad auctions. In particular, Bayesian versions of these models - the standard approach in auction theory - provide a very rich background for policy and mechanism design, and their learnability is not yet well-understood. For example, our study shows that all-pay auctions in a Bayesian setting with only two types *are* learnable, while they are *unlearnable* in the complete-information model with only a single type of bidder. Our analysis also covers Bayesian versions of the all-pay auctions but also with the war-of-attrition. [The latter has many similarities with all-pay auctions but it admits multiple Nash equilibria.]
> >
> > To build on your comment and the above, our revision zooms in on the differences between those games that are learnable and those that are not, and we analyze potentialness of these games. We think that this analysis adds interesting insights and provides further evidence for the usefulness of the potentialness as a metric of learnability.
> >
> > Now, as to your specific suggestion regarding stochastic games, we believe that this would be a very fruitful direction for future research, but any multi-stage model of interactions - stochastic, extensive, or dynamic games - would open up a completely new dimension that is beyond the scope of the current paper. We will of course be happy to discuss in detail the 2024 preprint by Guo et al. that you brought to our attention, but discussing the notion of potentialness in this context when no harmonic-potential decomposition exists for stochastic games in the first place would require a completely new paper in itself.
> >
> > Thank you again for your input. Kind regards,
> >
> > The authors

---

> > > ### Author Response · Authors · 2024-12-17
> > > **Rebuttal by Authors (3)**
> > >
> > > ### References
> > >
> > > - [BRMF+18] D. Balduzzi, S. Racaniere, J. Martens, J. Foerster, K. Tuyls, and T. Graepel, "The mechanics of $n$-player differentiable games," in ICML '18: Proceedings of the 35th International Conference on Machine Learning, 2018.
> > > - [CMOP11] O. Candogan, I. Menache, A. Ozdaglar, and P. A. Parrilo, "Flows and decompositions of games: Harmonic and potential games," Mathematics of Operations Research, vol. 36, no. 3, pp. 474–503, 2011.
> > > - [CMOP13] Candogan O, Ozdaglar A, Parrilo PA (2013) Near-Potential Games: Geometry and Dynamics. ACM Trans. Econ. Comput.
> > > - [GLZ24] X. Guo, X. Li, and Yufei Zhang, "An $\alpha$-potential game framework for ${N}$-player dynamic games, https://arxiv.org/abs/2403.16962
> > > - [LMP24] D. Legacci, P. Mertikopoulos, and B. S. R. Pradelski, "A geometric decomposition of finite games: Convergence vs. recurrence under exponential weights," in ICML '24: Proceedings of the 41st International Conference on Machine Learning, 2024. URL: https://arxiv.org/abs/2405.07224
> > > - [RR20] G. Ramponi, and M. Restelli, "Newton Optimization on Helmholtz Decomposition for Continuous Games"

---

### Decision · Action_Editor_8zDg · 2025-02-11

**Recommendation:** Accept with minor revision

**Comment:**

The paper introduces "potentialness" as a metric to assess how close a game is to being a potential game, aiming to understand the existence of pure Nash equilibria and convergence of no-regret learning algorithms. The study is well-motivated, clearly written, and empirically explores the behavior of potentialness across different game settings, including random and economic games.

After the rebuttal, two reviewers lean toward acceptance, while one remains skeptical about the work's overall contribution. The responses addressed several concerns, but the necessity and broader impact of potentialness as a metric remain open questions. Given the positive balance of opinions, the paper is recommended for Conditional Acceptance with revisions to clarify its significance, improve writing quality, and address computational concerns. Strengthening the justification for potentialness and exploring its broader applicability will enhance its impact.

**Audience:**

Yes, at least some individuals in TMLR’s audience would likely be interested in the findings of this paper. The study explores an empirically relevant metric for game theory and multi-agent learning, which aligns with topics of interest in theoretical machine learning. However, the extent of its impact may be limited by concerns over novelty and broader applicability, as some reviewers question whether the findings provide significant new insights beyond prior work. Strengthening the practical relevance and theoretical justification could increase its appeal to a wider audience.

**Claims And Evidence:**

The submission provides empirical evidence supporting the claims about potentialness and its correlation with Nash equilibria and learning dynamics. However, concerns remain regarding the broader significance of the metric and its novelty. While the mathematical derivations appear correct, the justification for why potentialness is a useful and practical metric beyond descriptive observations is not fully established. Clarifications on computational feasibility and relevance to broader game settings would strengthen the support for its claims.

---

> ### Author Response · Authors · 2025-03-11
> **Response by Authors**
>
> We're very happy to hear about the conditional acceptance of our paper. We carefully read through the paper again.
>
> Following your advice, we added text in the introduction to strengthen the justification for potentialness. In particular, we emphasize that wide-spread learning algorithms converge in many important economic games, but this cannot be explained by known conditions for convergence such as monotonicity or variational stability. Potentialness allows us to characterize games where no-regret algorithms converge and this will also be helpful in the analysis of algorithmic markets such as display ad auctions or oligopoly pricing where learning algorithms are widely used today.
>
> In our last revision, we added two paragraphs on page 8 to discuss scalability; furthermore, we improved writing quality throughout. We hope that this addresses the questions of the review team. After revising the paper once more, we hope that this version addresses all comments that remained.